# Interrupting Pedestrians in Indonesia; Effect of Climate on Perceived Steepness and Stair Climbing Behaviour

**DOI:** 10.3390/ijerph20010338

**Published:** 2022-12-26

**Authors:** Febriani F. Ekawati, Michael J. White, Frank F. Eves

**Affiliations:** 1School of Sport, Exercise and Rehabilitation Sciences, University of Birmingham, Birmingham B15 2TT, UK; 2Sport Coaching Education Program, Universitas Sebelas Maret, Surakarta 57126, Indonesia

**Keywords:** embodied perception, temperature, humidity, stair climbing, slant perception, sex differences, lifestyle physical activity

## Abstract

Increased activity during daily life is one public health initiative to reduce population inactivity. Increasing temperature and humidity influence walking for transport by reducing the blood supply available to exercising muscles. This study investigated effects of temperature and humidity on a perceptual cue, estimated stair slant, that can influence behaviour, and on subsequent speed of climbing. Participants (402 males, 423 females) estimated the slant of a 20.4° staircase at a university in Indonesia. Subsequently, the participants were timed covertly while climbing. As temperature and humidity increased, estimated stair slant became more exaggerated. Females estimated stair slant as steeper than males. For stair climbing, speed was reduced as temperature increased, and females climbed slower than males. Estimates of stair slant were not associated with speed of the subsequent climb. Climate influences estimates of stair slant that precede stair climbing and subsequent speed of the ascent. In this study, perception was unrelated to behaviour.

## 1. Introduction

Accumulation of physical activity during daily life is a current public health approach to counter inactivity in the developed world [1,2,3]. Increased stair climbing, rather than choice of escalators or elevators, is one initiative that can increase lifestyle activity for pedestrians [2,4,5,6]. Nonetheless, these interventions are not universally successful [7,8]. In Hong Kong, effects on population activity were minimal [9,10]. Climate appeared a barrier to the increased climbing target of public health [2]. This study tested effects of temperature and humidity on climbing behaviour and a visual cue that can deter activity, reported steepness of a climb.

Embodied approaches to cognition suggest that visual perception of a potential climb is influenced by the bodily resources available for the action. These resources include both bodily dimensions, consistent with Gibson’s affordance theory, and the energetic resources available to the climber [11,12]. Increasing temperature and humidity reduce the energetic resources available for physical activity by diverting blood supply from exercising muscles to the skin to regulate body temperature [9,10]. We tested whether the perceived steepness of a potential climb increased as temperature and humidity increased outdoors. Subsequently, we covertly measured climbing speed to provide a test of the effects of perception on behaviour that avoided effects of experimental demand [11,12]. We tested how a physical activity that public health would like to increase relates to the visual cue that we think prompts the behaviour in unconstrained pedestrians.

### 1.1. Climate and Resources for Pedestrian Behaviour

Climate influences physical activity [13,14,15,16,17], with sometimes an unexpected drop in the peak summer months [13,18,19]. Speed of walking is reduced at higher temperatures [20] and both participation and walking duration reduced with increased humidity [15,21]. As temperature and humidity increased in Hong Kong, climbing up an inclined travelator was less frequent [9,10].

The physiology of homiotherms underpins these effects on physical activity. The need to lose heat from exercising muscles increases blood flow to the skin [22]. Elevated temperatures impede surface heat loss. Redistribution of blood to the skin impairs delivery of oxygen to the muscles and, hence, reduces resources for climbing. When humidity and temperature increased, males were less likely to walk up the inclined travelator than females [10]. The greater muscle mass of males generates more heat that they must dissipate, disproportionately reducing climbing resources in males. The sexes differ in how they lose heat. Females have a greater surface area per unit mass and are better able to radiate heat from their skin than males who rely more on evaporative heat loss, i.e., sweating [22]. Air saturated with moisture, as it is when humidity is high, is a major barrier to sweating that explains greater effects of humidity in males than females [10].

Energetic resources for physical activity are limited [23]. As a result, pedestrians minimize the energetic cost of active transport to preserve resources [24,25,26]. Walking up slanted surfaces such as stairs or an inclined travelator requires more energy than walking on level ground. Climbing stairs required about three times the expenditure of walking speed that minimizes cost [27,28], and the inclined travelator in Hong Kong about twice more [8,28,29]. Raising all one’s body mass when climbing is always a vigorous physical activity. As a result, climbing is an energetically costly pedestrian behaviour that is generally avoided when an alternative means of ascent is available [30,31]. Perception of the steepness of the climb appears to be a visual cue that deters climbing [32].

### 1.2. Potential Effects of Climate on Perception

When pedestrians explicitly reported the angle of a slanted surface in degrees, a 23° staircase was estimated to be about 45° and a 5° hill to be about 20° [32,33]. In contrast, using the palm of an unseen hand to adjust a flat surface to match the slant of a staircase or hill, called a haptic measure, was more accurate [33,34,35,36,37,38]. Proffitt (2006) proposed that the available energy resources effected explicit angular estimates of a slanted surface [11]. He reasoned that exaggerated estimates of slant allow individuals to manage their energetic resources by modifying their behaviour [11].

If resources are depleted, explicit estimates of the steepness of a climb become more exaggerated. In experimental studies, individuals with reduced resources—temporarily, due to fatigue or the extra effort of carrying baggage, and longer term due to body weight, or age—explicitly estimate slants to be steeper than their comparison groups [32,33,35,38,39]. While experimental demand can never be excluded (e.g., Durgin et al. [40]; Durgin et al. [41]), quasi-experimental approaches can minimise its potential impact [38]. A preference for glucose [42] and higher amounts of body fat as deadweight to be carried [38], were associated with further exaggeration of staircase slant in situations where effects of experimental demand were unlikely. Participants waiting for a train completed a short interview about the station. Here, effects of natural variations in temperature and humidity on estimated angle represent a further quasi-experimental test of the effects of resources on slant perception.

Biological sex affects climbing resources, independent of effects on heat loss. On average, females have a greater proportion of their body weight as fat that must be carried yet have lower leg strength to do the carrying [23]. As a result, they have less climbing resources than males of the same weight and fitness [30]. Females consistently estimate slants as steeper than males [32,33,35,42,43] and are more likely to avoid climbing stairs when there is an alternative means of ascent [30,31,44]. Stair avoidance by pedestrians is a behaviour associated with perception of steepness [32,45]. Pedestrians who avoided climbing reported the slant of the stairs as steeper than their comparison groups. This effect occurred even when potential effects of individual differences in resources were controlled by stratified sampling and statistical adjustment [32,45].

This study investigated potential effects of climate on estimated slant of a staircase and climbing behaviour. We tested whether differences in temperature and humidity were associated with differential estimates of the steepness of stairs that would be subsequently climbed. We predicted that higher levels of temperature and humidity that reduce the blood supply to exercising muscles would be associated with more exaggerated estimates of the angle of the stairs but that there would be no effect on the haptic measure. We predicted greater effects in males than females (c.f., Eves et al., 2014) [32]. The vertical speed of stair ascent, measured covertly, was the natural behaviour of pedestrians climbing in different climatic conditions; faster climbs require more resource expenditure. Covert measurement aimed to avoid effects of experimental demand on the behaviour. We predicted reduced speed of climbing as temperature became less suitable for heat loss [20]. We predicted greater effects of climate on speed of climbing in males than females consistent with choice behaviour in Hong Kong [10]. In addition, we predicted slower climbing would be associated with more exaggerated estimates of stair angle.

## 2. Materials and Methods

### 2.1. Participants

Ethical approval was obtained from the Ethics Subcommittee of the University of Birmingham (ERN_15-0870). Eight hundred and forty-nine stair climbers at the Universitas Sebelas Maret were recruited for the study over 25 different days from 21 August to 30 September 2015. They were asked if they would volunteer for a study on perception of the environment. Once they agreed to be interviewed, they were asked for their estimates of the steepness of the stairs by the first author.

### 2.2. Stimuli

One set of stairs between buildings at the Medical School, Universitas Sebelas Maret, Surakarta, Indonesia was employed for the study (see Figure 1). Unlike the uniform staircases typical of the UK, this staircase started with three different sizes of tread depth, followed by twenty relatively uniform steps including two half-landings (3.93 m high, number of steps = 20; overall angle = 20.4°, angle of each section = 22.3°–24.2°). The stairs had open sides but were shielded from the sun above the heads of climbers. Therefore, air temperature and humidity could still affect the pedestrians.

### 2.3. Measures and Procedure

Participants faced directly at the stairs, standing 3 m away from the base of the 20-step section. They made two perceptual judgements, namely a verbal estimate of angle and a haptic measure, in a counterbalanced order of the 20-step section of the staircase. For the measures, participants were instructed to look straight ahead at the stairs and estimated the angle of the slant in degrees with respect to the horizontal, having been told 0° was horizontal and 90° vertical. For the haptic measure, participants used a Palm-Controlled Inclinometer [PCI; see Taylor-Covill & Eves (2013b) for more details [36]. Participants used the unseen palm of their hand to adjust a flat plate within the PCI enclosure until it paralleled the slope of the stairs. Following these judgements, participant’s sex was coded from appearance and they self-reported their age, height and weight. Height and weight were used to derive BMI of the participants. This interview took about 10 min. Finally, as the participants climbed to the top of the stairs, the experimenter measured their climbing time covertly using a stopwatch. Climbing was timed from when the leading leg was placed on the first step until both feet were placed on the top step. Weather data every hour, starting from 8:00 until 18:00 for that measurement day was obtained from Adi Soemarmo–Surakarta weather station. For each measurement day, air temperature (°C) and humidity (%) were recorded. During the study period of August to September 2015, temperature and humidity ranges were 24.8 °C–35.2 °C and 47–76%, respectively. Data collection started in the morning around 8:20 and finished early evening around 17:00. The data are available as Appendix A.

### 2.4. Statistical Analysis

Participants were excluded if they thought all staircases were 45° (*n* = 13), ran when climbing the stairs (*n* = 3) and paused while climbing the stairs to speak on the phone (*n* = 2). In addition, preliminary inspection with box-plots excluded two outliers on the angle estimate and four for the haptic measure. The final sample (*n* = 825) of 402 males and 423 females was relatively young (*M* age = 19.8 years, *SD* = 3.46) and of healthy weight (*M* BMI = 21.2 kg·m^−2^, *SD* = 3.20). Multiple regressions were used to test the effect of temperature and humidity independent of effects of sex, age and BMI. Unsurprisingly, temperature and humidity were highly correlated (Pearson’s *r* = −0.79, *p* < 0.001, 95% confidence interval [0.77, 0.82]); as the day warmed, humidity dropped. To counter this multi-collinearity between the two variables, humidity was used to predict temperature in the data set, and the residuals saved for inclusion in the analyses. The net outcome was a measure of temperature that was independent of humidity. There were differences between the sexes in BMI and the climatic variables (all *p* < 0.001), with a suggestion of a difference in age. Therefore, all variables were mean-centred within each sex to avoid confounding of the main effect of sex with the other variables. Multiple regressions were used to test the effect of climate and individual difference variables on (a) the angle and haptic estimates of staircase slant and (b) vertical climbing speed (m·s^−1^).

## 3. Results

### 3.1. Effect of Climate on Perception of Stair Slant

A preliminary multivariate repeated measures analysis of variance with sex as the between-subject factor and measure as the within subject factor revealed a main effect of sex (*F*_1, 823_ = 9.77, *p* = 0.002, *η_p_*^2^ = 0.012), a main effect of measure (*F*_1, 823_ = 1483.3, *p* < 0.001, *η_p_*^2^ = 0.643), and an interaction between the two (*F*_1, 823_ = 12.20, *p* = 0.001, *η_p_*^2^ = 0.015). While females estimated greater angles than males (Females = 39.1°, *SE* = 0.42; Males = 36.3°, *SE* = 0.43), there was no difference between the sexes for the haptic measure (Females = 23.3°, *SE* = 0.40; Males = 23.2°, *SE* = 0.41).

For the climate variables, the average temperature was 30.2 °C (*SE* = 0.09; range 25–35 °C) and the average humidity was 62.7% (*SE* = 0.25; range 47%–76%). Table 1 contains the standardized coefficients and summarizes the results of multiple regression analysis that included individual differences and climatic variables for the estimates of the stair angle. Inclusion of potential interactions of climate variables with sex revealed no interaction with humidity (*β* = 0.008, *p* = 0.852) and this term was dropped from the final model summarized in the table.

For the angle estimate, there were effects of sex in that females estimated the stairs as steeper than males, and older participants estimated the stairs as steeper, consistent with previous research [32]. For the effects of climate variables, there was an effect of humidity such that greater humidity was associated with steeper estimates. Further, the significant effect of temperature interacted with sex of the participant. Overall, the model explained 4.8% of the variance. For the haptic measure, a non-significant basic model, *F*_5, 819_ = 1.76, *p* = 0.124, explained no meaningful variance (0.4%) and is not presented.

Figure 2 and Figure 3 below summarize the effects of the climate variables on estimates of stair angle in males and females. The figures depict the values for one *SD* above and below the mean.

As can be seen from Figure 2, there were greater effects of increased temperature for males but not for females. Follow-up regressions for each sex separately revealed significant effects of temperature in males (*β* = 0.142, *p* = 0.005) but not in females (*β* = 0.031, *p* = 0.536). Inspection of Figure 3 reveals no evidence of an equivalent interaction with sex for humidity. Participants subjected to higher levels of humidity provided steeper estimates of stair slant in both sexes.

### 3.2. Effect of Climate on Behaviour

The average vertical climbing speed was 0.286 m·s^−1^ (*SD* = 0.034). A preliminary analysis revealed effects of the estimated angle on climbing speed (*β* = −0.086, *p* = 0.01) that became non-significant when sex was added (*β* = −0.051, *p* = 0.14) suggesting part of the effect of sex on speed below reflected differences in the estimated angle.

Table 2 summarizes the results of multiple regression analyses for the speed of climbing the stairs (m·s^−1^). There were significant effects of sex and temperature on climbing speed, with a significant regression equation that explained 6.4% of the variance. Both predictors were negatively associated with climbing speed. Females climbed the stairs slower than males (Females *M* = 0.279 m·s^−1^
*SD* = 0.033; Males *M* = 0.295 m·s^−1^
*SD* = 0.034). When the temperature went up, individual speeds were slower. There was no interaction between sex and temperature (*β* = 0.004, *p* = 0.93) and the term was dropped from the final model. Importantly, there was no significant effect of the slant estimate in the full model. In addition, there were no significant effects of age, BMI and humidity.

Figure 4 and Figure 5 below summarize the effects of the climate variables on speed of climbing. Once again, the figures depict the values for one *SD* above and below the mean to allow comparison with the effects of climate on perception. Follow-up analyses that tested for potential interactions between sex and the climate variables revealed no interaction between sex and temperature (*β* = 0.009, *p* = 0.850), despite the appearance of the figure, nor any interaction with humidity (*β* = 0.024, *p* = 0.612). Clearly visible in both figures is the faster climbing speed of males relative to females.

## 4. Discussion

As predicted, estimates of stair angle were more exaggerated as temperature and humidity increased. Consistent with previous work, females and older participants exaggerated the angle more than their comparison groups (c.f., Eves et al. [32]). There were no effects of self-reported weight and, as predicted, no effects of individual differences on the haptic measure. Speed of climbing reduced as temperature increased, consistent with walking on level ground [20] but, in contrast to Hong Kong, there were no effects of humidity. Females climbed slower than males as has been reported for walking on an inclined treadmill [46] and level ground [47,48]. Speed of climbing was not associated with estimates of stair angle in the final model.

### 4.1. Effects of Climate on Behaviour

The range for humidity in Hong Kong, 28–93%, was greater than here, 47–76% (c.f. Eves et al. [10]). Humidity, which disadvantages heat loss by sweating, is only an issue at higher saturations of air with moisture that impede sweating. Follow-up analyses of the Hong Kong data set for the truncated range of humidity in Indonesia were informative (see Appendix B for the full analyses). There were no effects of humidity on walking up the travelator in either males (Odds ratio [OR] = 1.015, 95% confidence interval (CI) = 0.994, 1.036, *p* = 0.173) or females (OR = 1.015, 95% CI = 0.997, 1.029, *p* = 0.121) in this restricted range. These null effects for humidity match the absence of effects on climbing speed for the Indonesian sample. For temperature, however, increases in Hong Kong reduced walking in males (OR = 0.849, 95% CI = 0.806, 894, *p* < 0.001) but had no effect in females (OR = 1.006, 95% CI = 0.968, 1.046, *p* = 0.750). For Indonesia, increases in temperature reduced climbing speed irrespective of sex. Climbing speed as a variable differs from the frequency of walking avoidance counted in Hong Kong and some differences are to be expected.

### 4.2. Effects of Climate on Estimates of Stair Slant

Proffitt [11] argued that explicit perception of slant, exemplified by estimates of the angle, allows pedestrians to choose a climbing speed that matches their available resources. Explicit estimates of slant have been linked to avoidance of climbing when an alternative method of ascent was available [30,45]. In the study here, higher levels of climate variables, that act as a barrier to lifestyle activity, were associated with enhanced exaggeration of the explicit perceptual signal that can deter behaviour. Both temperature and humidity influenced perceptual estimates whereas only temperature influenced subsequent climbing behaviour. Temperature had greater effects on estimated slant in males than females, echoing the follow-up analyses of the effects of temperature on behaviour in Hong Kong. These results are consistent with climatic effects on perception of a resource-related cue that *could* influence behaviour.

While preliminary bivariate analyses indicated effects of perception on climbing speed, estimated stair angle was unrelated to subsequent expenditure of resources when climbing in the final model. Individual morphology has major effects on the chosen speed that minimizes the energetic cost of walking [48,49,50]. For example, longer legs require faster optimal walking speeds [48,51,52] and, on average, male legs are longer than female ones. UK males also climbed faster than UK females on a university campus. Body shape, e.g., the distribution of mass on the leg, also affects optimal speed [48,49,50,51,52]. An individual’s morphology will affect chosen speed for all journeys made by that individual and, as such, is independent of effects of climate on a specific occasion that we report. Unlike cue-based avoidance of climbing, there was no alternative means of ascent here; the stairs were an unavoidable part of the journey being made by that pedestrian. Time pressure to complete a journey will influence speed, independent of any effects of resources on experienced slant. Individual morphology and time pressure could obscure any effects of resource-related perception on speed of climbing.

While Durgin and colleagues have argued, quite correctly, that explicit estimates could be influenced by experimental demand [40,41], the caveat is unlikely to be relevant to this quasi-experiment. There was no formal experiment (c.f., Taylor-Covill & Eves [38]). Instead, we tested the potential effects of natural variation in climate on perception. The pattern of effects such that there were greater effects of temperature on explicit estimates in males than females but no differential effects of humidity render demand an unlikely explanation. Effects of demand that vary differentially by separate climate variables seem implausible.

### 4.3. Strengths, Limitations and Future Directions

This research measured individual resource use when climbing stairs in unconstrained pedestrians for the first time. Climbing speed is a continuous measure of resource use by pedestrians, unlike the binary choice of avoidance investigated previously. Speed is better suited to regression analyses that contain other continuous measures. Covert measurement recorded natural pedestrian behaviour as a continuous variable. As a result, there was no experimental demand on the measure of behaviour. It should be noted that relating perception to unconstrained, real-world behaviour is very rare in this research field and represents a particular strength.

Inevitably on a university campus, the range for age and BMI was truncated and replication with a community sample would be more informative about those individual difference variables. Weight and height were self-reported, rather than measured, introducing error in BMI. Further, as an index of climbing resources BMI is imperfect; it contains both fat mass and fat free mass [32]. It is fat mass to be carried upwards that is important to perception and fat free mass has minimal effects [38]. While a measure of current physical activity levels might have been informative, behaviour was measured covertly and we did not wish to draw participants’ explicit attention to physical activity prior to climbing the second staircase.

A study with a climate chamber could attempt to confirm experimentally the effects reported here. Equivalent estimates of slant occur for life-sized displays of stairs and their real-world counterparts [37] and chosen walking speed on a slanted treadmill could index climbing behaviour. With such a design, effects of climate could be tested within subjects, controlling for effects of individual morphology on the behaviour throughout. Stratified sampling would be required.

## 5. Conclusions

As temperature increased speed of climbing stairs reduced, consistent with its effects for walking on level surfaces. Both increased temperature and humidity were associated with further exaggerations of reported steepness of a subsequent climb. Less suitable climatic conditions for heat loss influence a perceptual cue that is related to the resources used for physical activity. It is possible that the effects of climate on stair use are related to the perceptual signal for the climb.

## Figures and Tables

**Figure 1 ijerph-20-00338-f001:**
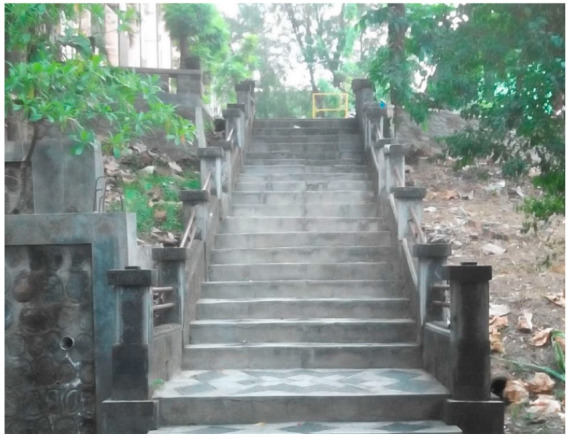
The outdoor staircase used for the study.

**Figure 2 ijerph-20-00338-f002:**
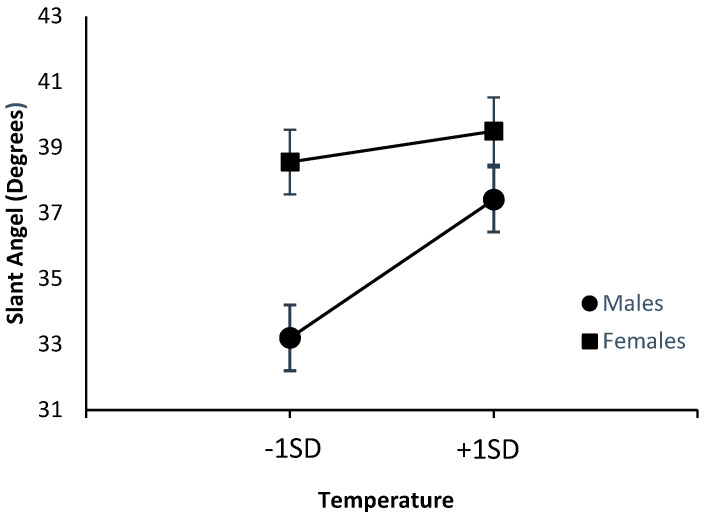
Effects of temperature on estimated slant angle.

**Figure 3 ijerph-20-00338-f003:**
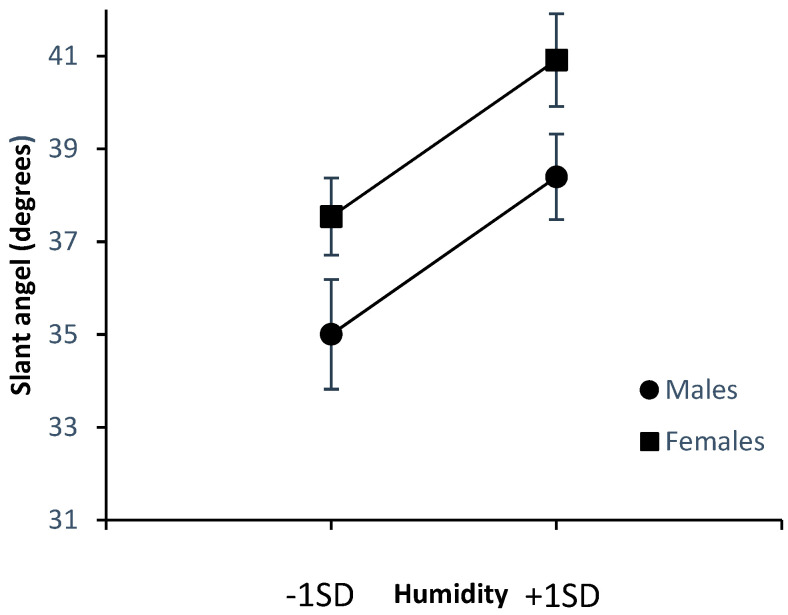
Effects of humidity on estimated slant angle.

**Figure 4 ijerph-20-00338-f004:**
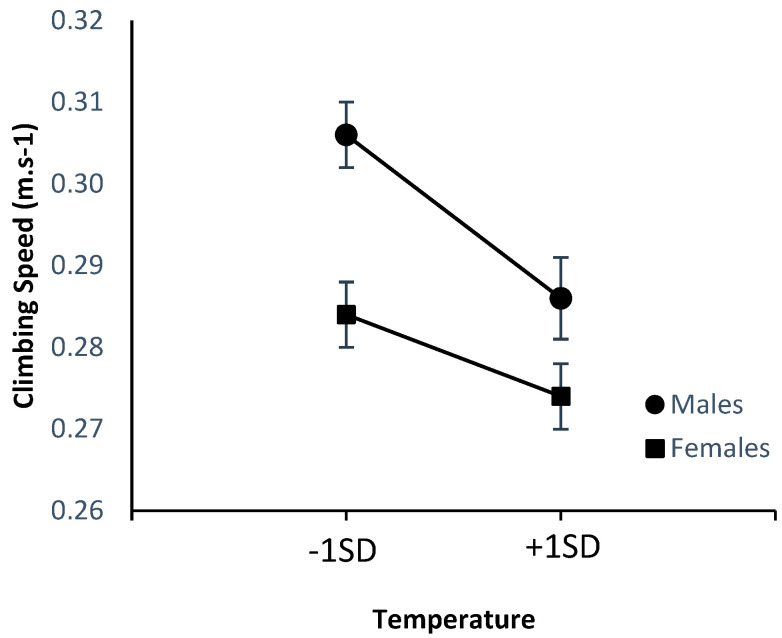
Effects of temperature on climbing speed (m·s^−1^).

**Figure 5 ijerph-20-00338-f005:**
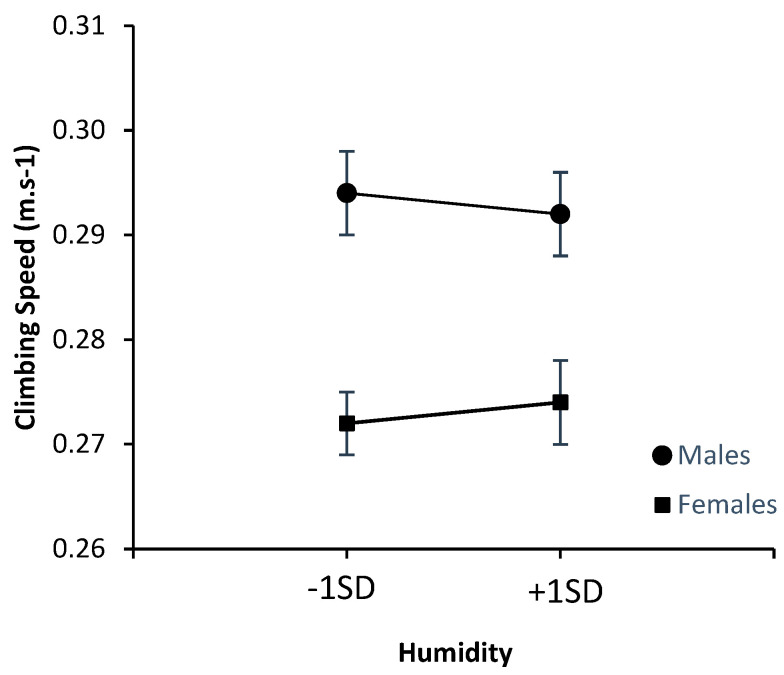
Effects of humidity on climbing speed (m·s^−1^).

**Table 1 ijerph-20-00338-t001:** Summary of the effects of individual differences and climate variables on estimates of stair slant in degrees.

Variable	Standardized Coefficients	95% CIs
Females > Males	0.161 ***	0.094, 0.228
Age, mean centred	0.125 ***	0.056, 0.194
BMI, mean centred	0.018	−0.050, 0.086
Temperature residual, mean centred	0.152 **	0.056, 0.249
Humidity, mean centred	0.076 *	0.009, 0.142
Sex × temperature interaction	−0.105 *	−0.201, −0.008
Adjusted *R*^2^	0.048	
*F* (6, 818)	8.01 ***	

Note: * *p* < 0.05; ** *p* < 0.01; *** *p* < 0.001.

**Table 2 ijerph-20-00338-t002:** Summary of the effects of individual differences and climate variables on vertical speed of stair climbing (m·s^−1^).

Variable	Standardized Coefficients	95% CIs
Females < Males	−0.229 ***	−0.295, −0.162
Age (years), mean centred	0.043	−0.026, 0.113
BMI, mean centred	−0.033	−0.101, 0.034
Temperature residual, mean centred	−0.104 **	−0.172, −0.037
Humidity, mean centred	0.022	−0.044, 0.089
Estimated angle, mean centred	−0.051	−0.118, 0.016
Adjusted *R*^2^	0.064	
*F* (6, 818)	10.37 ***	

Note: ** *p* < 0.01; *** *p* < 0.001.

## Data Availability

The data are available as Appendix A.

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
