# Peer review of "Interrupting Pedestrians in Indonesia; Effect of Climate on Perceived Steepness and Stair Climbing Behaviour"

_ijerph, 2022, doi:10.3390/ijerph20010338_

Round 1

Reviewer 1 Report

General remarks:

-Newell's constraints model (1986) can help you to conceptually give coherence to your problem and theoretical prevision to your results, because, as expected by the model, motor behavior emerges from the interaction between environmental, task, and intrinsic constraints.

- If perception is an important factor in your objectives, why didn't you discuss results based on percetion-action theories? 

-Why didn't you include a physical activity questionnaire (e.g., IPAQ)? We suggest you to state it in the limitations of the study. 

line 147- for correlations always present confidence intervals (CI) (mandatory)

Results- for inferential comparison tests always present effect size (mandatory)

lines 273, 295- review ortography

Reviewer 2 Report

Thank you for this article which was somewhat an interesting read but which requires significant improvement in order to be published. 

The title could be clearer and more concise. Who is interrupting pedestrians? There is no need for the semi-colon in the title.

Second line of abstract is unclear- "Walking for transport" "climate reduces resources for exercising muscles??

The statement of the problem is not explicit and the importance of this study or relevance are also unclear.

Line 54, what is a travelator? Do you mean escalator, elevator or the moving sidewalk present in the airport?

Methods section: is there any need to interview over 800 participants about their stair perception? Is there a reason why anonymous survey? Was not considered. It is unclear who interviewed the participants and for how long.

A picture of the staircase would be helpful.

Line 129: What is the purpose of self-reported height and weight and perceived sex by the researcher?

Strengths and limitation section: random statements are made without pointing out whether it is considering a strength or a limitation.

It would be helpful to point out the relevance of the study and to consider the protective nature of perceived steepness of the stairs in very high temperature for health.

Round 2

Reviewer 1 Report

lines 165-166- apply APA norms for CI presentation

Author Response

Corrected to APA style as requested.